# Synthetic mRNAs; Their Analogue Caps and Contribution to Disease

**DOI:** 10.3390/diseases9030057

**Published:** 2021-08-23

**Authors:** Anthony M. Kyriakopoulos, Peter A. McCullough

**Affiliations:** 1Nasco AD Biotechnology Laboratory, Sachtouti 11, 18536 Piraeus, Greece; 2Department of Internal Medicine, Division of Cardiology, Baylor University Medical Center, Dallas, TX 75246, USA; peteramccullough@gmail.com

**Keywords:** synthetic mRNA, analogue caps, elF4E, mTORC1, autophagy: immunity deregulation, maturation defects, autoimmunity, cancer

## Abstract

The structure of synthetic mRNAs as used in vaccination against cancer and infectious diseases contain specifically designed caps followed by sequences of the 5′ untranslated repeats of *β*-globin gene. The strategy for successful design of synthetic mRNAs by chemically modifying their caps aims to increase resistance to the enzymatic deccapping complex, offer a higher affinity for binding to the eukaryotic translation initiation factor 4E (elF4E) protein and enforce increased translation of their encoded proteins. However, the cellular homeostasis is finely balanced and obeys to specific laws of thermodynamics conferring balance between complexity and growth rate in evolution. An overwhelming and forced translation even under alarming conditions of the cell during a concurrent viral infection, or when molecular pathways are trying to circumvent precursor events that lead to autoimmunity and cancer, may cause the recipient cells to ignore their differential sensitivities which are essential for keeping normal conditions. The elF4E which is a powerful RNA regulon and a potent oncogene governing cell cycle progression and proliferation at a post-transcriptional level, may then be a great contributor to disease development. The mechanistic target of rapamycin (mTOR) axis manly inhibits the elF4E to proceed with mRNA translation but disturbance in fine balances between mTOR and elF4E action may provide a premature step towards oncogenesis, ignite pre-causal mechanisms of immune deregulation and cause maturation (aging) defects.

## 1. Introduction

Living cells and subsequently organisms obey specific laws of nature for existence [1]. Optimum ontogenesis, embryogenesis and morphogenesis function at growth rates of low entropy [1,2]. This low entropy offers the macromolecular structural existence that corresponds to an increase of degree of complexity that sustains life [2]. However, optimality is finely balanced. For example, in contrast to the life requirements and unexpectedly to researchers investigating the thermodynamics of analogue caps of synthetic mRNAs, the inter-molecular interactions between these caps with yeast [3] and mammalian [4], eukaryotic elongation factor 4E (elF4E) are at the same time both entropic and enthalpic. Complementary, the enthalpy of their inter-molecular reactions with elF4Es rises with the increase in the structural complexity of the synthetic caps [3,4].

The natural evolution’s complexity and diversification has provided the special quality control of mRNA capping to ensure for (a) an efficient protein translation (b) the stabilization of mRNAs from decomposition and (c) the normality of cellular functions [5]. The genetic control due to mRNA caps elevates in complexity with respect to the progress of evolution of species and is mostly evidently complex in eukaryotic mammalian species [5,6]. The main distinction between the different caps used for mRNA translation in higher eukaryotes is on the differential methylation of 7-methyl-quanosine-ppp (triphosphate linkage)-guanosine-5′ linkage present on the starting position of mRNAs. At the primary co-transcriptional and co-splicing events within the nucleus, a series of enzymatic activities by polymerase II retain the cap of mammalian mRNA unmethylated at the 2′O position, which is described as the m7GpppG (Cap O) structure [6]. The Cap O has the addition of a methyl group to the N7 amine of the quinine group by guanine N7 methylase, whereas further methylation of the first ribonucleotide at the 2′O position of the ribose by m7G-specific 2′O methyltransferase results to the m7GpppGm (Cap 1) structure [6]. Central to the differential methylation of mRNA caps is the Cap 2′O methyltransferase (CMTR1) enzyme, which adds the methyl group to the first nucleoside of the Cap 1 structure [6]. Further methylation by the CMTR2 [7] which adds an extra methyl group at the second nucleotide results in the m7GpppGmNm Cap 2 structure [7]. The enzyme activities of CMTR1 are within the nucleus whereas CMTR2 activities are both within the nucleus and the cytoplasm [7]. Complementary to the capping processes, the cytoplasm’s re-capping is a sophisticated self-preservation process of mRNA inactivation and reactivation regulation system of protein synthesis [5]. The re-capping of mRNAs is a frequent recycling event of mRNAs in mammalian cells [8], it is a highly frequent event in cancer cells [5] and helps to control and stabilize the cellular translation and stability of genome expression being a part of the transcriptome [9]. In addition to the stabilization of mRNA translation, the multiple mRNA methylations of mRNA caps have evolved as a part of the innate immune defense system [10]. The cap structures and capping processes become more and more specific for cells of different higher eukaryotic species and provide the first alarm for distinction of “self to non self” mRNA recognition under circumstances of cellular evasion of foreign genetic material as in the case of a viral infection, thereby offering essential signaling for interferon responses to recognize and encounter the viral infections [11]. Overall, the cap formations correspond to the regulation of mRNA translation under cell specific differential autonomy variables and signaling specificities.

Importantly, the numerous methylation sites on mRNA are associated to pathology conditions [6] and the mRNA caps are differentially methylated by nature’s evolution processes to circumvent cellular homeostatic imbalances [10,11]. Therefore, the investigation of potential associations between the analogue caps of synthetic mRNA caps, regularly used for the purpose of vaccination enhancement of stability and translation of immunizing proteins, and further the investigation of affiliations due to increased elF4E binding, with potential disease progression due to cellular hyper-sensitivities and cryptic pathologies that may be triggered, as well as the possible molecular pathogenesis pathways associated with clinical conditions becomes of medical interest.

In this respect, the out of cellular purpose continuation of phosphorylation capacity of the mTORC1 pathway is a possible candidate able to induce autoimmunity, cancer and other related pathologies and the molecular pathways involved are further underscored. Additionally, the possible pathology and clinical implications of pathogenesis induced by an mTOR deregulation, elF4E sensitivities and analogue caps (as well as other stability elements of synthetic mRNAs) in vaccines are hallmarked in this review.

## 2. The mRNA Caps

### 2.1. The Caps Used in Synthetic mRNA Formations for Vaccination Purposes

The overall design of mRNA genetic vaccine structures can be found in excellent reviews elsewhere [12,13]. An extensive summary of nature’s mRNA cap formations and synthetic cap structures used in construction of mRNAs in vaccines is presented in Table 1. During the almost five-decade history of mRNA vaccine development, which was initially destined to cure cancer [13], the restriction for use of the synthetic mRNAs was mainly due to their molecular instability, which has been resolved by the sequential experimental progress of the technology of capping. Historically, in the early 1970s, the quanytyltrasferase (GTPase) and the mRNA-quanine-7-methyltrasferase (Mtase) enzymes where isolated from vaccinia virions [14]. GTPase enzyme transfers GMP from GTP to 5′ terminus of unmethylated mRNA and sequentially the MTase transfers a methyl group from S-adenosylmethionine to the position 7 of quanosine to form the Cap O structure [5]. However, during the natural cap processes, about one third of mRNA remains uncapped. This is due to the addition of m7GpppG in the reversed orientation so that the m7G nucleotide instead of constituting the cap becomes the first transcribed nucleotide [5]. In fact, the researchers Pasquinelli et al. 1995 [15] have never observed methylated dinucleotides of m7GpppGm being incorporated in RNA in the normal orientation. In addition, the reverse capped mRNAs apart from attracting an antibody response from antibodies specific to the m7GpppG structure [15] are only scarcely translated [8]. The research data on the reverse capped small nuclear RNA (snRNA) reveal a clear-cut deficiency for nuclear import due to the inability of reverse capped RNAs to form RNA—protein (RNP) complexes and for this reason they become hypermethylated to form m-2-2-7G caps; this contrasts the mRNAs that have caps with normal orientation [5,8,9,15]. Further, the reverse capping of RNAs creates a cellular translational disturbance as the snRNAs form dynamic RNA-RNA complexes that catalytically intervene with the normal spliceosomal activities of nuclear pre-mRNA maturation events [16].

The vaccinia capping complex, however, although seemingly functional in nature’s processes, has not been used by the industry to overcome capping problems by reverse added orientation due to additional enzymatic step requirements that may have complicated large-scale production [12]. Instead, the discovery of anti-reverse-caps (ARCAs), [17] which are chemically synthesized and added to the 5′ end of mRNA favored their industrial use since they contribute to a 2.3 to 2.6 fold higher protein translation than normal. Indeed, the finding of increases in protein translation has been made ever since the ARCA structures as the cornerstone of synthetic mRNA production [12,13]. The ARCA structures are chemically synthesized caps and are made to possess a substitution at 3′0–OH of 7′ methyl guanine ribose with hydrogen (H+) or –OCH3 moieties constituting respectively the 7-methyl (3′O-deoxy) GpppG and 7-methyl (3′-O-methyl) GpppG capping structures [17]. These chemical adaptations of mRNA caps apart from stabilizing the synthetic mRNA “deceive” the eukaryotic initiation complex and the deccapping complex [5] to favorably forward the translation of synthetic mRNAs. Complementary to the ARCAs the synthetic capping technology has devised structures of caps that have insertions of moieties like the imidodiphospate within the tri or tetra-phosphate bridges of ARCA structures to yield the NH, CH2, CCl2 and CF2 analogs [18]. These cap structures show an even more increase in resistance to the enzymatic decapping complex and also a higher binding affinity to eukaryotic translation initiation factor 4E (elF4E) protein (Table 1) [18]. Moreover, the insertions of 1,2-dithiodiphosphate within the phosphate bridges of ARCAs yield the SH analogs which show even higher affinities for binding to the eIF4E [19].

The general principle on the progress of devising caps for synthetic mRNAs focus on the chemical modifications of the caps that will confer, when added to the synthetic mRNAs, (a) a decrease of the susceptibility to degradation of the synthetic mRNA by the decapping complex and (b) an increase to the binding efficiency of the capped synthetic mRNA to the eIF4E. Furthermore, in order to bypass the cellular translation restriction of 2′O methylation and therefore the rejection of synthetic mRNAs as “non self” mRNA, a 2′-O-methyltransferase capping enzyme is used to methylate ARCA caps of synthetic mRNAs and denote the “self” Cap 1 structure to be recognized adequately by the recipient cellular translation machinery [20]. This may mean however, that even under problematic conditions of homeostatic imbalance [6,10,13] seen during a concurrent viral infection [13], the recipient cells will be forced to translate the synthetic mRNAs due to the capping they possess, even if their requirements for keeping cellular homeostasis are different. Notably, in the natural processes, any cap modification has its own physiologic consequences, and these arise due to the influence of the affinity of the cap to various cofactors involved in the specific translator machinery of gene regulation [21]. Furthermore, in the higher eukaryotic organization of protein translation, the mRNA capping enzymes have evolved to become sensitized by specific regulatory signals that sequentially correspond to (a) their change of behavior in terms of genome expression, (b) their differential localization within the compartments of the cell, and (c) their divergence of activities [21]. These sensitizations eventually determine the overall gene expression and have a direct impact on the normality of the cell functions and thus on the cell’s fate [21]. In Table 1, the naturally existing caps for mRNAs and the chemically modified analogue caps for synthetic mRNAs are listed.
diseases-09-00057-t001_Table 1Table 1The naturally existing caps of mRNAs and the synthesized caps used for mRNA vaccination.A/The most common natural quanosine methylated mRNA cap modifications and their biological functions Name of cap modifications and enzymes involvedStructure and use by different species Site of capping and biological function Cap O: sequential methylation of the first quanosine nucleotide [5,6]RNA TPaseGTaseguanine-N7 MTasem7G(5′)ppp(5′)GUniversal for all eukaryotic mRNAsUsed by most virusesNuclear mRNA capping Recruitment of pre-mRNA protein complex for splicing, polyadenylation and nuclear export.Protection from nonsense mediated decay.Efficient nuclear export by cap binding complex (CBC) CytoplasmProtection from exonuclease cleavageAffix of elF4E-p to assemble the elF4F complex for initiation of translation. Regulation of gene expression by CBC and elF4E-p.Cap N6A: substitution of first transcribed guanosine nucleotide by adenosine methylated at 6N position [20,21,22]Multi component protein complex consisting of catalytic subunit Methyltransferase Like 3 (METTL3)m7G(5′)ppp(5′)AmpNp 20–50% of m7G(5′)ppp(5′)Xm mRNA caps in Hela cellsCommon to human and mouse mRNAUsed by selected viruses including Influenza A virus (IAV), HCV, HBV, HIV, Simian virus 40 (CV40), and enterovirus 71Co-transcriptional modification.Control of mRNA splicing.Depriving decapping activity.Transcription start site (TSS) signaling. Epitranscriptomic gene regulation.Regulation of viral infection and host immune response. Promotion of RNA Decay.Cap independent mRNA translation.Cap 1: Methylation of the +1 ribonucleotide at the 2′O position of the ribose [6]m7G-specific 2′O methyltransferase (2′O MTase) cap methyltrasnferase 1 (CMTR1)m7G(5′)ppp(5′)GmLower and higher eukaryotes (mouse and human)Selected virusesEukaryotes: nuclear co-transcriptional modification. Restriction of Cap O dependent initiation of translation “as non self” during cellular evasion of foreign mRNA. Promotion of antiviral response by induction of interferon stimulated gene (ISG) proteins.Type 1 interferon signaling leads to expression of interferon-induced proteins with tetratricopeptide repeats (IFIT).Binding of IFIT 1 to Cap O instead of elF4E-p attenuates mRNA translation and replication of viruses that do not encode their own 2′O Mtase, e.g., SARS-CoV, West Nile virus.Viruses: post transcriptional modification at cytoplasm. Evasion of recognition by the innate immune response of the host and interferon response by viruses using a) host mRNA capping pathways (conventional capping) e.g., herpesviruses and retroviruses or (b) viral (non-conventional) e.g., coronaviruses and paramyxoviruses expressing their own 2′O Mtase as in case of SARS-CoV-2. Cap 2: Methylation of the +2 ribonucleotide at the 2′O position of the ribose [7]CMTR-2m7G(5′)ppp(5′)GmNmHigher eukaryotes (human)Nuclear and cytoplasmic Independent from N7 methylation of guanosine (Cap O) or from methylation of first nucleotide 2′O ribose (Cap 1). Greater affiliation for cap2 methylation in Cap 1 mRNAs.Used for efficient pre-mRNA splicing (small nuclear RNAs).Promotion of Cap 1 dependent mRNA restriction of translation as “non self” against foreign (viral) mRNA.Cap NAD+: 5′ end NAD+ [9]Bacterial RNA polymerase (RNAP)Eukaryotic RNAP IISensitive to nuclear migration protein nudC (containing NAD-capped RNA hydrolase)Sensitive to DXO degradation proteins in humanNAD(5′)pNpBacteriaYeastsHumanNuclear during initiation of transcription, -3- 4-fold increase in mRNA stability in bacteria. Non canonical initiating nucleotide (NCIN)-mediated initiation of transcription.Mammalian cells are equipped with a distinct NAD+ capping mechanism from transcription initiation that does not support cap-dependent translation. Combination of de-NADing proteins in human control the mRNA decay mechanisms and the gene expression.**B/The synthetic mRNA capping systems**Biochemical synthesis pathways Structure Biological properties 5′ terminal mRNA modificationby vaccinia virus enzymes [14]Guanylyl transferase and S adenosylmethionine: mRNA (guanine-7) methyltransferaseG(5′)ppp(5′)Gp G(5′)ppp(5′)Apunder presence of S adenosylmethionine m7G(5′)ppp(5′)Gmp m7G(5′)ppp(5′)AmpPost-transcriptional modificationNo sequence specificity apart from terminal purine. Acting by function of poly(A) as a substrate for all methylation reactions. Anti-reverse cap analogs (ARCA) [17]Biochemical modifications using pyrophosphate bond formation reactions7 methyl(3-deoxy) GpppG7 methyl(3-O-methyl)GpppG7 methyl(3-O-methyl)GpppG m7G(5′)ppp(5′)GInhibit reverse capping of m7GpppGm by bacteriophage polymerases. Increase 2.3-2.6 fold of mRNA translation compared to m7GpppG cap in a rabbit reticulocyte lysate systemIncrease protein translation in a dose dependant manner in EL4 cancer cells In combination with optimum poly(A) length and β-globin 3′utranslated regions significantly improve RNA stability in immature dendritic cells (human). Capping enzyme system and 2′O-methyltrasferase to generate Cap 1 [20] 7-methylGpppGmContribution to enhancement of transgene expression in stimulated T cells (human) in combination with optimized UTRs and poly(A) lengths. Equivalent in translation efficiency to the ARCA capping system.Imidodiphosphate: NH analogues, [18]Methylenebisphosphonate:CH2 analogues, [18]Dihalogenmethylenebisphosphonate: CCl2 and CF2 analogues, [18] Biochemical moiety substitution within the 5’,5’-tri- or tetraphosphate bridge of mRNA caps (including ARCAs)…5′pNHp5′……5′pCH2p5′……5′pCl2p5′……5′pCF2p5′…Sequential increase in binding affinity to the elF4EDecrease degradation by DcpS (decapping scavenger in exosome) and decapping complex Dcp1–Dcp2 Cap analogues with 1,2-dithiodiphosphate moieties [19]Biochemical dithiodiphosphate modified nucleotide synthesis applied to ARCA synthesis2S analogs: phosphorothioate (O-to-S) substitution inside the triphosphate bridge2S ARCA analogs: modified by the presence of a 2′-O-methyl group 7,2′-O-dimethylGppSpGD17,2′-O-dimethylGppspsGD1D27,2′-O-dimethylGpppspsGD1D2Dramatic increase in ElF4E-p binding affinity Decrease of decapping susceptibility by SpDcp1/2 enzyme complex Increasing efficiency of mRNA translation in immature dendritic cells (D1 to D1D2)D1: gold standard for the treatment of melanoma 

### 2.2. The mRNA Capping during Normal and Oncogenic Conditions

The initial isolation and identification of GTPase and Mtase enzymes from vaccinia virus grown in Hela cells [14] was fundamental for the capping technology synthesis and the final manufacturing of mRNAs as vaccines. The identification of these enzymes was probably feasible to achieve due to the regular epigenetic burst vastly present in cancer cells [5]. However, and particularly in Hela cells, the abundant N6-methyladenosine (m6A) modifications of mRNA by m6A methyltransferases induce the expression of Snail mRNA in a cap independent manner [21,22,23], as the modifications of mRNA by the m6A methyltransferases constitute a dominant epitranscriptomic level of gene regulation [18]. The switching of the cap dependent to the cap independent mode of translation is often for cells under stress conditions. The reasoning that the Cap O and Cap 1 structures universally predominate in all cell functions appears to have been misguiding for many years, as it has been shown by recent advances of NAD+ cap identification and mRNA decay surveillance mechanisms [22]. The frequent presence of O’ 2 methylation on the ribose of the second transcribed nucleotide “the so-called Cap 2 formation”, and the first nucleotide, adenosine N-6 methylation, prognosticate for the differential translation efficiency in different cell lineages and under specific stimuli [18,19]. In the experimentation by Holtkamp S et al. 2006, quite rightly the authors conclude that an evident enhanced RNA stability and translational efficiency was devoted to the presence of globin untranslated regions (UTRs) and also to the concurrent presence of a 120-length poly A tail incorporated to the synthetic mRNAs and not to the presence of the Cap O structures that had been inserted to the tested synthetic mRNAs [24]. In this experimentation, [24] which is considered as fundamental for the optimization of synthetic mRNA technology, [12,13] the detection of optimum synthetic mRNA expression was measured by means of antigen presentation on EL4 mouse malignant T-cell lymphoma cells. This can also be a misguidance as these cancer cells are often used in these kinds of experimentations because they stably express fluorescent proteins and are thus a useful experimental tool for cancer biology research than may not reflect normal conditions (personal communication). In respect to the protein translation efficiency, the overwhelming research data from investigations of mRNA translation processes that cells follow during normal conditions and conditions of oncogenesis, reveal the presence of differential sensitivities of caps and the elF4E binding protein that further dictate the differential expression of numerous sets of genes. For example, the messenger RNAs by genes required for normal cellular functions like the glyceraldehyde 3-phosphate dehydrogenase (GAPDH), and the *β*-actin, are less sensitive to elF4E activity as compared to mRNAs from oncogenesis genes involved in cell growth proliferation and immune responses such as the C-MYC, the Bcl-2, the vascular endothelial growth factor (VEGF), the cyclins and others [25]. In particular, in nature, the elF4E sensitivity is attributed to the 5′UTR structures located near the caps that dictate the cellular choices for a cap dependent and/or independent mRNA translation rather than the caps themselves [26]. Moreover, the excessive presence of elF4E concentrations above the required threshold during normal conditions is vital for the proper cellular functions and maintenance of homeostasis. Furthermore, during the burst of reactive oxygen species (oxidative stress) which is a hallmark of oncogenesis, the evolutionary guided excess of elF4E protein which otherwise is present ubiquitously during normal conditions is consumed to promote the cancerous cell transformations and the survival of malignant cells [27].

Furthermore, the methylation of caps is also important for disease onset [5,6]. In a gene-specific manner proto-oncogenes like c-myc induce the 5 ‘guanosine cap methylation in order to promote cellular proliferation [28]. By this way, the target genes of c-myc (*elF4E*, *elF4A1, elF2B1* and others) are induced in their expression constituting a cascade of auto-induction between methylation of caps and expression of target growth factor genes. Once bound consistently, the methylated caps on elF4F complex are attractants to signaling proteins as the ribosomal signaling scaffold proteins and like the receptor for activated C Kinase 1 (RACK1), that thereafter, promote the translation of other short mRNAs and its activity is increasing in medical importance [29].

### 2.3. The elF4E Dependence Can Be a Provocation to Commencement of Disease

The elF4E protein has independent roles in the nuclear export of mRNA and its cytoplasmic translation and nevertheless in most cases if not all the elF4E protein expression is a powerful regulon and in most conditions is regarded as a potent oncogene governing cell cycle progression and proliferation at a post-transcriptional level [30]. The unregulated binding affinity of elF4E has profound physiological consequences that can lead to a disease. For example, the elF4E efficient binding to the mRNA caps recruits the elF4G and elF4A proteins to form the elF4F binding complex. Once the elF4F complex becomes intact, the translation of mRNAs, especially those mRNAs that are equipped with the highly stable structures of 5′ UTRs stabilizing the *β*-globin gene translation [31] and which are fundamentally used in the manufacturing of synthetic mRNAs used for vaccination [24], becomes highly efficient [12,13,24]. The increased binding affinity of caps used in synthetic mRNAs [17,18,19,20], and thereafter, the increase in translation efficiency of synthetic mRNAs of vaccines means that the elF4E is rendered to become more readily bound to the elF4F complex for a prolonged time than normal and also that may avoid forming complex with the 4E binding proteins (4EBPs). Since the synthetic mRNAs with their analogue caps are specially designed to increase overwhelmingly the translation of their encoded sequences [17,19] this will also prolong the existence of their attractive 5′ UTR structures to other endogenous mRNAs that contribute to gene regulation [26]. In addition, the highly translation regulated mRNA of cyclin D1 depends on its translation rate on the unwinding of the elF4E structure by the helicase elF4A [26] which will be readily abundant during increased translation of synthetic mRNAs [5,10,21,25]. The elF4E protein is a powerful regulatory protein of cap dependent translation [25]. When elF4E becomes bound to elF4G, the activity of elF4A is liberated and this is independent to the elF4E cap binding activity [25] and also when elF4E is bound to elF4G its auto-inhibitory functions are counteracted. However, elF4E, whether unbound or bound to the elF4G complex, stimulates translation by more than one distinct mechanism of translation and the translation of sensitive mRNAs (those that are capped) is increased with elF4E activity [21,25]. In this sense, mRNAs that are involved in growth, proliferation, transformation and differentiation of cells will be preferentially stimulated during an induced increased cap dependent protein translation activity by the readily bound analogue caps. In this respect, although the determination of increased translation by analogue caps modification has been performed in highly proliferative malignant cell lines [18,19], the chemically modified cap methylation of mRNAs is shown to elevate pro-oncogene expression, and vice versa, the cap methylation is being promoted in malignancy [32]. Figure 1 illustrates pathways that lead to disease by prolonged phosphorylation of elF4E serine 209.

For many important reasons, the elF4E is appraised as an oncogene [33,34]. It has long been postulated that the activated elF4F complex induces a deregulation of phosphorylation between elF4E and the 4E binding protein 1 (4EBP-1) that leads to tumorigenesis [35]. On the other hand, this is the reason that the inhibition of unbound or bound forms of elF4E on the cap initiation complex (m7GTP–eIF4E interaction), and therefore, the inhibition of cap-dependent translation, is considered as a potent anti-tumor therapeutic target [35]. It is due to the above that the 4EBPs (1, 2 and 3) sequestering elF4E have been assessed for their anti-oncogenic potential [34]. However, the abundance of phosphorylation between the mechanistic target of rapamycin (mTOR) protein kinase and 4EBPs causing release of BP-1 from elF4E do not identify clearly the actual biologic role of the 4EBPs. The data obtained by one 4EBP may not apply to other 4EBP and research investigations accumulate data declaring distinct cell-type specificities with respect to the role of 4EBPs [34]. Although both 4EBP1 and 4EBP2 are modestly decomposed by phosphorylation and are released from the elF4E binding and set elF4E free to promote cap dependent initialization of translation, and these are ubiquitously expressed in all mammalian cells, their amounts present are significantly different between organs [35]. For example, the 4EBP-1 is abundant in the skeletal, muscle, adipose tissue and the pancreas, whereas the 4EBP-2 is present in higher amounts in the central nervous system [35]. Remarkably, the activity of p53 proto-oncogene inhibitor activation on the diminishment of protein translation by de-phosphorylation of 4EBP-1 leads to the subsequent attenuation of elF4E cap binding translation via the mTOR pathway [36].

The over expression and phosphorylation of 4EBP-1and 4EBP-2 is encountered in many cancers [36] and systemic autoimmunity conditions [37]. As the mTOR kinase is also a major coordinator of the T helper cells differentiation and regulates their cellular fate decisions, the loss of mTOR control and specifically the loss of the mTORC1 dependent pathways can lead to the disorganization of protein synthesis and T cell dysfunction predisposing to immune irregularities [37]. In particular, through the p13/AKT/mTORC1 pathway as illustrated in Figure 1, when the 4EBP is over expressed, it becomes hyperphosphorylated and is thus incapable of controlling mRNA expression [34]. On the other hand, the depletion of 4EBP1 expression is oncostatic against tumor progression via the AKT dependent translation pathway [38].

### 2.4. The elF4E Serine 209 Phosphorylation Is Important for Disease Onset

Whilst the 4EBPs are phosphorylated by the mTORC1 kinases, the elF4E is phosphorylated in the irreplaceable site of serine 209 (human sequence) by the bonded mitogen actine protein kinases (MARK), kinases MNK1 and MNK2 [34]. This double way of 4EBP and elF4E phosphorylation results in the dissociation of phosphorylated 4EBP thereon called p-4EBP, from the 4EBP-ElF4E complex and thus the initiation of translation by the phosphorylated elF4E (p-ElF4E) and the assembly of the elF4F complex is triggered [34,35,36].

Thus, so far, the actual role of serine 209 phosphorylation of elF4E remained controversial [34]. However, recent experimental evidence supports the notion of p-elF4E dependent translation in MYC and ATF 4 drives oncogenic initiation progression and aggressiveness of cancer in a rate-limited manner [39]. Latest scientific evidence [31], suggests that the p-elF4E maintains the 4EBP-elF4E binding [34] (Figure 1), and by using the most ancient arm of integrated stress response (ISR) in mammals and the general control of nonderepressible 2 (GCN2), the p-ElF4E maintains the AKT/4EBP-1 signaling and the stress response by ATF4 transcription factor. In this process, the mTORC1 activity is maintained to phosphorylate 4EBP-1 and thus the anti-oncogenic potential of mTOR silencing is inactivated [27]. In this respect, any excess of cellular mRNA translation, as is in the case of overloaded synthetic capped mRNAs in vaccines, and especially when present in initiative and progressive conditions of autoimmunity [37] and oncogenesis [38,39], the surplus drive of the p-elF4E-cap-dependant translation is far more than desirable for the maintenance of cellular homeostasis. It must be re-emphasized that synthetic mRNAs are designed to have solid capping structures and UTRs to ensure for efficient and long-lasting translation [10,12,13,17,18,19]. Thus, an elevated cap-dependent protein translation under circumstances of PI3K/AKT pathway activation which causes *MYC* and *ELF4E* genome amplifications [40], and active translation of synthetic m RNA vaccine bearing caps with increased affinity to elF4E than normal [10,11,12,13,17,18,20], may drive even more the potential of cells for autoimmunity and oncogenesis. This is especially more important as the mammalian species have evolved mechanisms of elF4E surplus in the cell for their normal development [27]. For example, the translation of ferritin heavy chain 1 (Fth1) is highly sensitive to elF4E expression levels in a dose dependent manner [27]. Moreover, the synthetic mRNAs are modified structurally to carry the 5′ untranslated repeats (URTs) of β-globin gene in order to confer translational efficiency and stability [12,13,17,20,24]. The above-normal limits regular presence of 5′ UTRs and the overwhelming attractiveness by the long-lasting presence of p-elF4E through the regular binding of iron-responsive element binding protein/iron regulatory protein (IRE/IRP) regulatory network, is said to drive the activation of nuclear erythroid-derived 2-like 2 (NRF2) transcription factor targeting the gene of FTH1 [41,42,43,44]. According to specific cellular vulnerabilities [41], this causes a disturbance in the cellular iron availability and in the regulation of ferroptosis of the cells. As the analogue caps of the synthetic mRNAs are designed to increase binding to elF4E and the translation cycles are increased considerably [17,18,19,20], this can make accessible the 5′ UTRs for longer than normal and predispose to activation of NRF-2 as illustrated in Figure 1.

## 3. The Loss of mTOR Functions

### 3.1. The Origin of mTOR and Its Links to Disease

The mechanistic target of rapamycin, or otherwise mTOR, is a kinase enzyme namely, the FK506-binding protein 12-rapamycin-associated protein 1 (FRAP1) [45]. The mTOR is a pivotal regulator for cell growth, metabolism, and aging, firstly identified in yeasts in 1991 [45,46,47]. At the cellular level, mTOR is involved in (a) the regulation of transcription and translation, (b) the control and stability of a plethora of mRNA transcripts, and (c) the actin cytoskeletal organization, all activities by performing epigenetic regulation through regulation of autophagy [48,49,50,51]. In the yeast cells, there are two mTORs, namely the Tor1p (tor1), and the Torp2 (tor2), which are the mammalian orthologs of mTOR1 and mTOR2 [49]. The tor1 assembles solely into TORC1 and tor2 assembles preferentially to the TORC2 large protein complexes. These complexes are also conserved in mammalian species and human. The mTORC1 complex contains both tor1 and tor2. The mTOR as a whole structure, interacts with diverse cell signaling pathways, and stimulate protein synthesis through phosphorylation of crucial translation regulators, such as the ribosomal S6 kinase which is the major protein S6 kinase (S6K) in mammalians [49,50,51], and the 4EBPs (both 1, 2, and 3 4EBPs). The phosphorylation of S6K reinforces the mRNA translation. Furthermore, as the ribosomal proteins and other regulators related to the reinforced translation by the S6K also regulate the encoding of related genes of these translated mRNAs, the whole process constitutes a signaling cascade [52,53]. In this respect, the activated S6K1 phosphorylates the MDM2 inhibitor protein and thus indirectly inhibits the nuclear localization and ubiquination of the tumor suppressor p53, through the p38α/AKT/mTORC1/S6K1 pathway [54]. Loss of p53 function promotes the cellular proliferation via mTORC1 activity inhibiting autophagy and cell cycle arrest as illustrated in Figure 2. The other important targets of mTOR are the 4EBPs, and specifically the 4EBP1, which binds to and inhibits the eIF4E from binding to the 5′end caps of eukaryotic mRNA transcripts as illustrated in Figure 1 [55]. The phosphorylation of 4EBP1 by mTOR subsequently results in the separation of the 4EBP1 from the eIF4E [55]. As a result, the surplus of liberated eIF4E released from binding to the 4EBP1 enhances the translation of mRNAs (in a rate-limited manner) of the key growth-promoting proteins, the cyclin D1, the c-Myc, and the VEGF as illustrated in Figure 1 [25,26,33]. The regulation of cell growth is mediated by mTOR through two of the key extracellular and intracellular signaling factors (a) the growth factors (insulin and insulin-like) and (b) the nutrients such as amino acids and glucose [56,57].

The dysfunction and the deregulated signaling of mTORs are implicated in metabolic, neurodegenerative, and inflammatory disorders and malignancy [47,48]. In general, the mTOR deregulation is strongly associated with tumorigenesis. As mTOR inhibits autophagy under normal cellular conditions, its deregulation increases cell proliferation instead of driving cells to normal death as it is illustrated in Figure 2 [46]. By the loss of function of autophagy regulation by mTOR (whereby inhibition of mTOR the autophagy is induced) [58], and once the autophagic cell death is reduced, the activation of oncogenes is then enhanced [42]. The induction of autophagic cell death (or otherwise known as Type 2 cell death) is tightly connected to: (a) the expression of Beclin-1 tumor suppressor gene (present in many cancers), (b) the induction of death-associated protein kinase (DAPK) expression (which otherwise is oncostatic and is being reduced by the restriction of autophagy), and (c) the expression of the dual phosphatase and tensin homolog deleted on chromosome 10 (PTEN), which is also downregulated during the inhibition of autophagy [58,59]. Nevertheless, another important enzymatic activity comes from the domain of the death associated protein kinase 1 (DAPK1) which is a member of the family of serine/threonine kinases. The association of DAPK1 with the tuberous sclerosis 2 (TSC2) interferes with the binding between TSC1 and TCS2 by inhibiting TSC2 and thus the ability of TSC1 to halt the signaling activity of mTOR is abolished [60], as illustrated in Figure 2. On the other hand, however, the DAPK induces phosphorylation within the bona fide 3 (BH3) domain of Beclin1 protein and thus promotes the dissociation of Beclin-1 from Bcl-2 and BclX(L) inhibitor which are liberated from complex, a case that leads to the induction of autophagy [61] as illustrated in Figure 2. Notably, the attenuated elF4E is tightly connected with decreased Bcl-2 activity independent from Akt activation signaling in breast cancer constituting elF4E a potential target for therapy [62]. Although at present, the knowledge about mTOR and the Beclin-1 and/or DAPK, and the loss of autophagy regulation is limited [60], it is clinically important to consider the deregulation of mTOR to autophagy and its implications in the pathogenesis of a series of tumors and neurodegenerative disorders as a result of an external stimulus. For example, it is strongly hypothesized that Huntington disease and the spinocerebellar ataxias are related to the mTOR activity without having yet a clear mechanism been identified [46].

### 3.2. Hamartomas and mTOR Deregulation

Hamartomas are mainly benign tumors caused by mutations in several tumor-suppressor genes and can provide a fine example of mTOR deregulation. The hamartoma syndromes are divided into two categories [46]. The first category is directly associated with the dysregulation of mTOR and the second category, which is clinically similar to the first, has also its eventual causality and pathogenesis attributed to the mTOR deregulation [46]. The tuberous sclerosis complex (TSC), the PTEN-related hamartoma and the Peutz-Jeghers syndromes belong in the first category. Moreover, Lhermitte-Duclos disease, Proteus syndrome, Cowden disease and the Bannayan-Riley-Ruvalcaba syndrome are all autosomal-dominant benign tumors linked to the TSC pathway of pathogenesis. The tumor-suppressor gene of phosphatidylinositol 3,4,5-trisphosphate 3-phosphatase (PTEN) is mutated in all of these four pathologies. Furthermore, the already mentioned TSC1 and TSC2 protein complexes are the signaling regulators for the expression of mTOR. When these dysfunctions or loose activity and/or when the over-expression of mTOR coincides, these are the key pathogenesis events that lead to TSC disease [63,64]. The neurofibromatosis type 1 (NF1) disease leads to malignancy when the rat sarcoma (RAS) suppressor gene NF1 that encodes neurofibromin is homozygote mutated [65]. The RAS hyperactivation activates subsequently the AKT/mTOR pathway along with the Raf/MEK/ERK pathway leading to inhibition of autophagy, increase of cellular proliferation, and subsequent development of the NF1 syndrome [66,67]. The Von Hippel-Lindau syndrome (VHL) causes multiple tumors in numerous organs. Mutations of the Von Hippel-Lindau Tumor suppressor gene cause loss of the products that otherwise inhibit the ubiquitylation and the proteasomal degradation of the hypoxia-inducible factor-1-α (HIF-1α) [68], a case that involves numerous nuclear translocatio ns and intranuclear hypoxia dependent signals. As mTOR is an active positive regulator of HIF gene products, it is also strongly implicated in the causality of VHL syndrome [46,69,70].

The representative syndromes that belong to the second category of hamartomas are the family adenomatous polyposis (FAP) and juvenile polyposis syndrome (JPS). FAP is caused by germline mutations in the adenomatous polyposis coli (APC) gene that results in the triggering of upper and lower gastrointestinal polyps and carcinomas, hepatocellular carcinoma and hepatoblastoma (less frequently) [71]. Further, the activation of β-catenin signaling is linked to the loss of APC function that leads to promotion of cellular survival and inhibition of cell death [71]. The *β*–catenin works through the Wnt/β-catenin pathway to promote cancer [72]. This is important for mTORC1 deregulation since the PI3K/AKT/mTORC1 pathway works closely with the Wnt/*β*-catenin pathway regulating one another (thus, constituting a common therapeutic target against the colorectal cancers that they produce), and therefore, the activating signals of mTOR are considered as predisposing conditions for FAP development [46,72,73]. The JPS which is also characterized by multiple polyps along the gastrointestinal tract has predominantly (to a 50–60% of patients) germline mutations in *SMAD4* or *BMPR1A* genes [74]. Although mTOR has not yet been identified as a contributor to the development of cancer from JPS mutations, there is interlinking evidence that Bmpr1a phosphorylation regulates Smad activation in response to Bmp and associatively the Bmp activates the cellular differentiation of osteoblast through Smad4 and mTORC1 pathways [75]. Table 2 lists the mTOR malfunctioning related disorders in relation to pathways providing sensitivity to elF4E.

### 3.3. The mTOR and the Immune System Regulation

The mTOR has a central role in the mammalian immune response and metabolism [71]. The mTOR activity, by connecting the extracellular with the intracellular environment, contributes to the adaptive immune system regulation, having a main role in the cell differentiation, maturation, migration, and antigen-presentation [67,78]. The metabolic pathways are strongly related to the immune pathways, and mTOR is a regulator in various immune lineages such as the mast cells, neutrophils, natural killer cells, γδ T cells, macrophages, dendritic cells (DCs), T and B cells. The upstream and downstream diverse signaling of mTOR, regulates and interact with numerous key factors such as growth factors and nutrients [48].

The mTORC1 signaling enhancement induces the T cell activation and reversibly the mTORC1 inhibition during immune response, which promotes T cell anergy. This pathway is essential for performing cancer immunotherapy since tumors evade the immune surveillance defense mechanisms, and in this respect, the activation of mTORC1, can constitute a target for anticancer therapy [79]. In addition, mTORC1 is involved in the T cell maturation. The research teams of Pollizzi et al., 2016 and Verbist et al., 2016 provide scientific evidence showing that the mTORC1 activity is elevated in the “effector-like” daughter cells and is reduced in the “memory-like” daughter cells [80,81]. The Class I phosphoinositide-3 kinases (P13K) are important cellular signaling mediators inducing cell growth and proliferation with the main representative being the p110α protein which is involved actively in oncogenesis [82]. Notably, as the cytokine and the growth factor receptors induce the activation of mTOR, the CD28 precisely promotes the activation of PI3K and subsequently the mTORC1 in T cells [73,78]. Further, in this respect, the interleukin 2 and 4 (IL2 and IL4), activate the mTORC1 pathway through the activation of PI3K [73,78,83,84]. Moreover, the stimulation of mTOR to produce IL-12 and IFN-γ prolong further the activation of mTOR in CD8 + T cells whereas the production of IL-1 induces the generation of Th17 type of immune response through the stimulation of mTOR [78,85]. The Th17 T cells are a distinct subtype of CD 4+ cells holding a unique type of immune response, the Th17 type of immune response, which holds a central role in contributing to the pathogenesis of autoimmune and inflammatory disorders [78,83,86].

### 3.4. The Wnt Signaling Pathway, the mTOR, and the Immune Regulation

The Wnt (name that arises from the name of Drosophila gene wingless and the name of a vertebrate homolog gene int-1), constitutes an evolutionary ancient signaling pathway formatted by growth stimulated proteins and encoded by the highly conserved wnt gene from lower vertebrates to human [87]. Wnt is a fundamental signaling pathway that has, as protagonists, a plethora of secreted glycoproteins that follow complex cascades to determine the cells fate, migration, polarity, and organogenesis in an early embryonic life [84,87]. When the wnt signaling pathway works in close relation with the β-catenin protein, it is termed as the wnt/b-catenin dependent pathway, or otherwise when β-catenin protein is not involved it is termed as the wnt/β-catenin independent pathway. However, there are yet more wnt signaling pathways not associated with or without the β-catenin and mTOR seems to be implicated in at least one of them [57]. As long as the TCS1/TSC2 tumor suppressing complex remains intact, the mTORC1 remains inactive and protein synthesis remains halted (Figure 2). However, there are cases, where, and through wnt inactivation of GSK3β kinase (that regulates the silencing of the mTORC1 activity especially under conditions of energy deprivation and hypoxia), the GSK3β inhibition is bypassed and wnt pathway activates mTORC1 independently from the kinase Akt kinase and ribosome 6K kinase inhibitory activities. These cases constitute an important step leading to many cancers and neoplasms directly via the wnt activation of mTORC1 as illustrated in Figure 2 [88]. Notably, mTOR is also activated potently by amino acid induction, independently from the wnt pathway [57].

Under the presence of suitable conditions such as under presence of oxygen, amino acids, growth factors and lipids, the mTOR remains in its active form and induces translation, lipid synthesis, and mitochondrial biogenesis [55,63]. However, by the lack of these conditions, the mTOR signaling is inhibited and thereby autophagy is enhanced, creating a feedback loop. The conversion of a resting T cell to an active T cell demands nutrients and active glycolysis reactions. This activation pathway is potently related to the Akt/PI3K axis. Respectively, the CD28 promotes the PI3K activation resulting to Akt activation, thereby resulting to the inhibition of lymphocyte metabolism that inhibits in turn the T cells from functioning properly [89]. Nevertheless, CD28 is known as the major stimulation pathway for naive T-cell activation, and the CD28/B7 co-stimulatory complex is mostly relevant in the combat of immune defense against clinically challenging conditions [89]. During CD28 dependent co-stimulatory signaling, the Akt kinase is repressed since the levels of ATP are high and hence the low levels of AMP do not allow AMP kinase to phosphorylate TSC2 and thereafter repress mTORC1 as illustrated in Figure 2 [89,90].

In many ways, the role of mTOR in T cell differentiation and in relation to CD28 co-stimulatory signaling remains problematic [78,90]. However, the CD28 dependent pathway is synergizing with the T cell receptor (TCR) and the IL-2 synthesis and the inhibition of CD28 and IL2b result in low mTORC1 activity, ineffective clonal differentiation, and cellular anergy [59]. Importantly, the mTOR-deficient T cells can not differentiate to Th1, Th7, or Th2 effector cells in vivo or in vitro under potent polarization [81]. The environmental factors in this respect play a central role in the activation of mTOR and consequent metabolic and immunologic activation that lead to T cell differentiation (as the mTOR controls numerous transcription factors that regulate T cell differentiation by antigen recognition) that become important [60]. The mTOR phosphorylation through ribosomal S6K [85] is induced by antigen recognition in CD8+ T cells and through the MAP-kinase signaling. Complementary, the development of memory cells is depended on the expression of IL-7 (CD127) and IL-2 receptor beta chain (CD122) during a contraction phase [61]. These CD8+ T cells act as effectors and mTOR contributes to the generation of these stimulators. In this way, the inhibition of mTOR activity leads to the loss of CD8+ memory T cells activities. As the above activities of mTOR (which are related to various factors) show the dependence of numerous immune cell functions to the mTOR activation or inhibition, the destabilization of normality functions of mTOR may lead to immune dysregulation. In most senses, when the molecular breaks of mTOR activity are deregulated, the orchestra of the immune system contributors is deregulated too, leading the immune cells to become prone in developing reactions that may lead to inflammation, autoimmunity, and tumorigenesis as illustrated in Figure 1.

### 3.5. Other Clinically Important Implications Arising by mTOR Improper Signaling

The role of mTOR in the normal brain’s functioning is essential since mTOR proper functioning contributes to the development and feeding of the brain neurons and the circuit formation. The mTORC1 activity inducing the mRNA translation near the synapses of brain neurons is vital for the neuronal system development. Notably, the deregulation and overexpression of mTORC1 is high in patients with TSC and this is accompanied by epilepsy (90%), autistic features (50%), and benign brain tumors [84,91].

As autophagy is interlinked with induction of the mTORC1 signaling [46,47], mTORC1 plays also a key role in the glucose homeostasis in the liver. Specifically in the fasting periods when the glucose intake is deprived, mTORC1 is under functioning due to the inhibition of AMP kinase dependent and independent pathways [78]. Moreover, not unexpectedly, the mTORC1 is involved in the control of hepatic ketone body accumulation. During low glucose presence, the mTORC1 activity is reduced and so ketogenesis becomes under functioning. However, when mTORC1 signaling is persistently activated, as for example in mice with an activated TSC1 due to persistent mTORC1-dependent suppression of peroxisome proliferator-activated receptor alpha (PPAR-α), autophagy is impeded, and as a consequence, (a) amino acids are not produced and (b) gluconeogenesis is blocked. This failed response to gluconeogenesis due to mTORC1 deregulation results in fatal hypoglycemia [92]. Furthermore mTOR, activity is vital for the β-cells of Langerhans islets in the endocrine region of pancreas, as mTOR is a central positive regulator of β-cell growth, proliferation and function [49,50]. In mice, the mTORC1 deregulation contributes to reduced blood glucose levels, induces glucose intolerance, and therefore, hyperinsulinemia. This subsequently induces the constitutive increase of the mass and number of β-cells (increase in their proliferation and production). In this regard, mTORC1 is a regulator of β-cells mass and function. Further, the action of mTORC1 in mice is depended on the S6K1 and by the absence of S6K1 the size of β-cells becomes smaller, the cells become glucose-intolerant with an increased insulin resistance due to an impaired insulin secretion. By the chronic insulin resistance, the pancreatic β-cells react with a compensating mechanism. The constitutive insulin resistance elevates the pressure in β-cells of Langerhans islet and after a period of time the β-cells become overloaded. This compensates with the observed β-cell hypertrophy, the increased β-cell proliferation, and subsequent formation of progenitor cells due to the attempt of compensation and due to the increased demands of insulin production and secretion. The aforementioned condition, if not reversible in a given time, leads to type 2 diabetes condition due to the steady insulin resistance. In the mice being genetically obese or in the high-fat-fed mice, mTORC1 activity is increased. The axis mTORC1/S6K1 has a key signaling role in the regulation of glucose homeostasis. If this axis remains activated over a time threshold, the pancreatic β-cells lose their mass through apoptosis and due to the feedback inhibition of IRS1 and IRS2, which promotes cell-death. Because of the sustained cellular apoptosis, the pancreas loses β-cells, a condition leading to β-cell compensated hypertrophy and elevated insulin resistance, hyperinsulinemia and hyperglycemia [50,51]. In this regard, the known antidiabetic drug metformin, can improve the glucose homeostasis through the partial inhibition of mTORC1 signaling.

The neurodegenerative disorders (a) Parkinson’s disease, (b) multiple sclerosis, and (c) Alzheimer disease are also associated with the hyperactivity of mTORC1 [48]. A central shared pathway of this disease group is the accumulation of toxic proteins, the so-called inclusion phenomenon, which constitutes the failure of neurons to clear mutated or misfolded proteins that subsequently causes cell death. The autophagy and the ubiquitin-proteasome system, which are avoided due to the mTORC1 hyperactivity, are the main pathways in the intracellular protein degradation system. Centrally, the defects associated with the autophagy activation contribute to the pathogenesis of neurodegenerative disorders in most cases. For example, another pathological pathway of neurodegeneration observed in mice, is the deletions in autophagy genes Atg5 and/or Atg7, which lead to the aggregation of polyubiquitinated proteins without a necessary presence of any disease associated protein mutation [92,93,94]. In this regard, autophagy is a pivotal survival mechanism of the neurons and its impairment results in neurodegeneration. In most senses, the normal functioning of mTORC1 is fundamental for the neural homeostatic balance.

Research studies on the role of rapamycin as an inhibitor of mTORC1 activity have shown that the inhibition of mTORC1 impairs cellular aging and subsequently prevents age-related diseases such as cancer, neurogenerative disorders, type 2 diabetes and stem cell dysfunction pathologies. Therefore, the mTORC1 is also regarded as key negative regulator of cellular lifespan and its false signaling activation causes aging pathologies. In aging mice, the mTORC1 signaling pathway is elevated in the hematopoietic stem cells (HSC) [95] and the activation of mTORC1 by Tsc1 loss causes premature aging in HSC. Furthermore, the inhibition of mTORC1 signaling, for instance through rapamycin targeting, promotes HSC renewal and improves hematopoietic function, by improving the longevity of cells [86]. Finally, the mTORC1-S6K1 and mTORC1-4E-BP1 axes have an important role in the signaling pathways which can cause a diverse range of pathogeneses. In this respect, the inhibition of these two axes can possibly expand the life expectancy of cells [78]. The mTORC1 may also activate complementary mechanisms without having an association with protein synthesis. Nevertheless, the autophagy activation linked to mTORC1 inhibition, may contribute to some of the effects of cellular extension of life span. Notably, as the mTORC1-4E-BP1 axis manly inhibits the elF4E to proceed with mRNA translation, defaults in fine balances between mTOR and elF4E action can constitute a premature step of oncogenesis and ignite pre-causal mechanisms that can lead to stem cell related disorders and aging defects [85,86,89].

## 4. Conclusions

The deregulation of phosphorylation between elF4E and the 4EBP-1 leads to tumorigenesis. Although for many years the role of serine 209 phosphorylation of elF4E remained controversial, it has been recently elucidated that p-elF4E supports the notion of mTORC1 maintenance of activation by using the most ancient arm of integrated stress response (ISR) in mammals, the general control of nonderepressible 2 proteins. Therefore, the dependence to the elF4E and further to the whole of el4F complex and is constituents becomes important for disease onset. Natural capping of mRNAs has substantial differences in complexity and methylations as compared to the analogue caps synthesized to increase bounding to the elF4E, promote translation, and decrease natural chances of decapping processes within the cell. This will prolong the existence of analogue caps in translation machinery of cells longer than normal. Therefore, special attention must be cared off as during mTORC1 deregulation the necessary pauses for increase cellular proliferation, cell growth and induction of oncogenes drive to conditions of disease. Conditions of elF4E and mTORC1 irregularities are of medical importance. The analogue caps of the synthetic mRNAs used for vaccination against cancer, genetic disorder therapy and nowadays as emerging for infectious diseases are optimized to stabilize and increase the translation of the encoded proteins in mRNAs and this is done to provide an efficient immunization. In this respect, attention should be made on studies that have shown that the enthalpic increase and entropic change between synthetic cap interactions with mammalian elF4E as well as with the elF4E of lower eukaryotic species may be in contrast to the elevation of complexity of living organisms in terms of growth rate requirements and compatibility with health (and life). However, our analysis reflects the potentiality to disease onsets by elF4E and mTOR deregulations. As the internal variables of a living organism are trying to keep its internal state unchangeable (homeostasis), and the reactions between analogue caps and elF4E are thermodynamically not favorable [1,2,3,4], this has to be analyzed by further superior physical chemistry, biochemistry and explicit toxicity evaluation research. Particularly during sensitive circumstances, as during deregulation of fine balance of cellular homeostasis (conditions of elF4E and mTORC1 deregulation) and as of this consequence, due to the autophagy deregulation, this is said to cause immune dysfunction irregularities, cellular maturation incompatibilities and predispose to various autoimmune disorders and malignancies. Foremost attention must be made to the potentiality of the loss of cap regulating innate defense of cells. By the alteration of regulation of cap methylation, this sets the organism susceptible to viral and bacterial infections as well as other diseases.

## Figures and Tables

**Figure 1 diseases-09-00057-f001:**
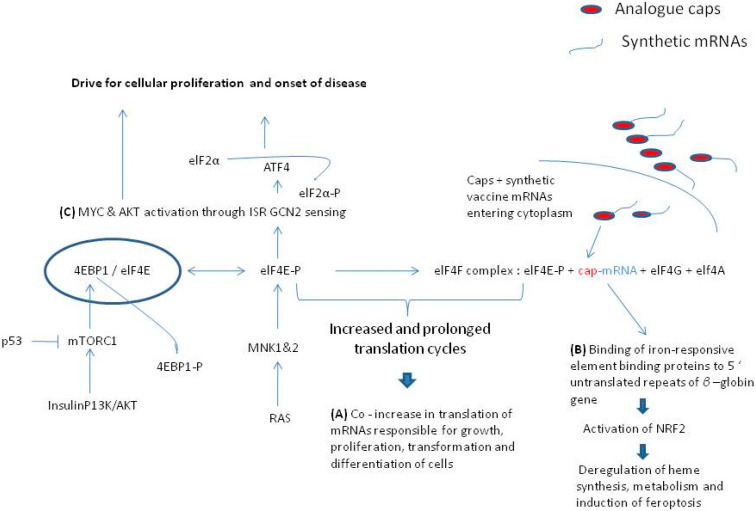
Pathways that lead to disease from the prolonged phosphorylation of elF4E 209 serine. (**A**) Intense cap binding to the p-elF4E and elf4G complex facilitates the co-translation of cellular proliferation proteins (C-MYC, cyclin D1, Bcl-2 and others) [25,26]. (**B**) Prolonged presence of 5′ untranslated repeats of *β*-globin gene of mRNAs are attractant to the activation of NRF 2 protein that induces ferroptosis [41,42,43,44]. (**C**) Prolonged presence of phosphorylated form of elF4E (p-elF4E) is said to activate ISR GCN2 sensing and drive cellular proliferation to induce auto-immunity and oncogenesis [31,34,36,37,38,39,40].

**Figure 2 diseases-09-00057-f002:**
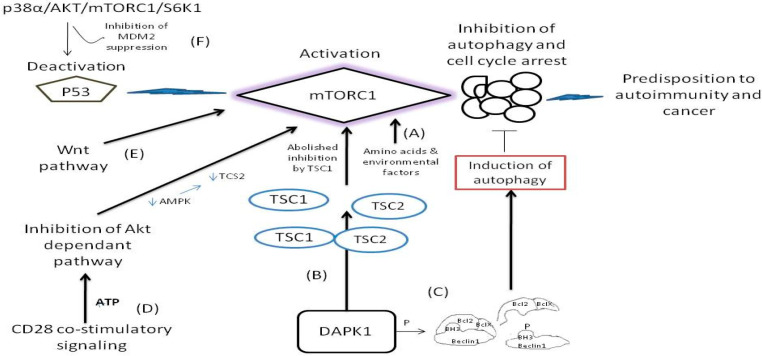
The pathways where mTORC1 remains activated or its function is bypassed and predispose to disease. (**A**) The mTOR remains activated by insulin and insulin like molecules and nutrients such as glucose, amino acids and other environmental factors [48,56,57]. (**B**) Interaction of DAPK1 with TSC1 causes inhibition of mTORC1 suppression. Following, the liberated TSC2 from TSC1/TSC2 complex activates mTOR [58,59,60]. (**C**) Phosphorylation of Beclin 1 in its BH3 domain causes dissociation of Beclin 1 from its inhibitors Bcl-X(L) and Bcl-2 to activate autophagy [61]. (**D**) During CD 28 signaling the Akt kinase activity is repressed therefore alleviating mTORC1 inhibition [89,90]. (**E**) Wnt pathway activates directly the mTORC1 bypassing GSK3β inhibition [88]. (F) Deactivation of p53 function pause breaks for mTROC1 activity [54].

**Table 2 diseases-09-00057-t002:** Disorders related to loss of function of mTOR activity.

Benign Tumors	Ways of Sensitivity to elF4E
Hamartomas	PTEN/P13K/AKT pathways [46,76]
**Benign disorders that lead to malignancy**	
Neurofibromatosis type 1	AKT/Raf/MEK/ERK pathways [65,66,67]
Von Hippel-Lindau syndrome	Hypoxia inducible factor 1 [66,67,77]
Family adenomatous polyposis	Wnt/β-catenin signaling P13K/AKT [68]
Juvenile polyposis syndrome	Smad 4 pathway [75]

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
