# Peer review of "Synthetic mRNAs; Their Analogue Caps and Contribution to Disease"

_diseases, 2021, doi:10.3390/diseases9030057_

Round 1
Reviewer 1 Report
The authors reviewed recent advance of research on regulation of synthetic mRNAs used in vaccination against cancer and infectious diseases. This is timely review and should be informative to many researchers. Although mTOR inhibits the elF4E to proceed with mRNA translation, I feel that most of the contents in “3. The role of mTOR” are not directly related to the topic.
Minor:
Lane 22: Manly: Mainly?
Lines 268-275, 383-384. Figure legend should be located under the figure.
Author Response
Dear Reviewer please seea attached the reply to your comments.
Sincerely
Dr Anthony M. Kyriakopoulos BSc(Hons), MSc, MD/Ph.D.FIBMS

Reviewer 2 Report
This manuscript describes the effects that p-elF4E shows on the enzyme mTORC1 that regulates the cell growth, cell signaling pathways and protein synthesis trhough phosphorylation regulators. It gives an extensive view about the mechanism affected by mTOR and the effects that elF4E (a powerful RNA regulon and potent oncogene agent) shows in the development of diseases.
The introduction section sounds more like a book chapter than a scientific journal review. The first paragraph does not grab readers' attention, so I suggest to authors to modify it.
Figure 1 is a bit blurred, please modify it to make it more clear.
It would be good to have a quick look for readers if authors make a table that describes different disorders that are related to mTOR activity and which of them are more sensitive to be altered by elF4E.
Author Response
Dear reviewer
Please see attached our reply to your comments.
Sincerely
Dr Anthony M. kyriakopoulos BSc(Hons), MSc, MD/Ph.D.FIBMS

Reviewer 3 Report
Review for the manuscript diseases-1323918
The manuscript provides a review on literature of mRNA capping structures comparing natural occurring ones and the synthetic analogues used in therapeutic approaches (e.g. vaccination). Further it reviews literature of known pathways, mostly from oncology, on the correlations of affinity for binding of proteins (in particular eIF4E) as initial step for protein translation and their relative translation ratios across the endogenous genes.
The summary of capping strategies and comparison of natural vs synthetic strategies is interesting. There is a sound literature for the correlation of the mTOR and mTORC, WNT-pathway in oncogenesis. This parts are well prepared and based on references.
The reviewer does however not see sufficient literature evidence of synthetic caps with their high eIF4E affinity inducing sufficient shift in translation of oncogenes (VEGF, Cyclin D1, BCL-2) to support the presentation as a review. Evidence for the interdependencies depicted in Figure 1 should be clearly related to specific research papers showing such effect in order to publish such review paper. In current form it is still quite speculative.
Transient shifts in translation upon an external trigger are experienced all the time in human cells. The reviewer would also like to ask the authors to address the effect of duration of synthetic capped mRNA presence/application on the possibility of irreversible shifts in the translation pattern, which might cause ultimately a risk on oncogenesis or other diseases.
Also Figure 2 would need a caption phrasing main correlations between pathways and citing the respective research papers showing that.
With current hints that synthetic mRNA caps may cause under certain circumstances, the reviewer finds it inappropriate to cause diffuse alarm for the field on many possible disease scenarios (oncogenesis, autoimmunity, neurodegenerative diseases) and publish it as review paper. In the conclusion most statements are written in conjunctive (may be, seems to, can cause, may prone), thus the manuscript is to hypothetic/speculative for a review paper.
Providing a clear evidence for where the authors exactly see highest concern(s) (e.g. long-tern or repeated application; application in szenario of predisposition, etc) and would like research gaps to be filled, could turn the manuscript in an opinion paper or perspective and being more beneficial to the research community.
Author Response
Dear reviewer
Please see attached our reply to your comments.
Sincerely
Dr Anthony M. Kyriakopoulos BSc(Hons), MSc, MD/Ph.D. FIBMS

Round 2
Reviewer 3 Report
“Although many studies have correlated deregulation of protein biosynthesis with cancer, it remains to be established whether this process is necessary and/or sufficient for neoplastic transformation and metastasis.” Citing Signaling control of mRNA translation in cancer pathogenesis. EC Holland, N Sonenberg, PP Pandolfi and G Thomas; Oncogene volume 23, pages 3138–3144 (2004).
Thus, introducing synthetic mRNA with higher affinity for eIF4F and stability, it remains to be established whether that is supportive or even in some cases sufficient or for neoplastic transformation or initiation of autoimmune disease.
Modifications of synthetic mRNA, including but not limited to the cap structure, affect not exclusively the affinity to translation initiation factors but also the mRNA stability and interaction with recognition molecules of the innate immune system. Observed increase in translation is often a combination of these factors. Biochemical assays allowing dissecting these effects from each other are an option, but so far rather scarce performed.
I searched but could not find a single research article clearly showing the effect of synthetic caped mRNA on enhanced expression of proto/oncogenes or cell transformation. Working since several years on mRNA delivery using mRNA with synthetic caps, we observed the need for repeated treatment due to the transient expression of the transgene, but never any sign of transformation or prolonged cell life for transfected cells.
Several viruses also tune eIF4F binding but not directly are sufficient to be tumorigenic, due to the multiple control mechanisms our cells use to secure translational control (tumor suppressor genes, microRNAs, feedback-loops, stress response and immune system sensors). Nature has developed many feedback loops and control mechanisms to avoid that a single event can have detrimental effects in higher organisms.
Thus, without any direct evidence that synthetic mRNA caps are sufficient to induce cell growth transformation or autoimmunity, for this reviewer the manuscript remains in its major message - that synthetic mRNA caps are a potential cause of disease – at the status of a hypothesis. Also after the changes in the manuscript revision, my opinion is that this manuscript should be better published as a comment, opinion, or future perspective paper. This could motivate research in the according field to investigate in detail, while a review might be seen as evidence for a causal connection not yet demonstrated (despite around 10 years of research on mRNA as therapeutic option and several clinical trials).